# Signal-sensing triggers the shutdown of HemKR, regulating heme and iron metabolism in the spirochete *Leptospira biflexa*

Juan Andrés Imelio[1], Felipe Trajtenberg[1], Sonia Mondino[1], Leticia Zarantonelli[2], Iakov Vitrenko[3], Laure Lemée[3], Thomas Cokelaer[3,4], Mathieu Picardeau[5], Alejandro Buschiazzo[1,6]*

1 Laboratory of Molecular & Structural Microbiology, Institut Pasteur de Montevideo, Montevideo, Uruguay, 2 Unidad Mixta Pasteur INIA, Institut Pasteur de Montevideo, Montevideo, Uruguay, 3 Plateforme Technologique Biomics, C2RT, Institut Pasteur, Université Paris Cité, Paris, France, 4 Bioinformatics and Biostatistics Hub, Institut Pasteur, Paris, France, 5 Biology of Spirochetes Unit, Institut Pasteur, Université Paris Cité, Paris, France, 6 Dept of Microbiology, Institut Pasteur, Université Paris Cité, Paris, France

* alebus@pasteur.edu.uy

**Data Availability Statement:** All relevant data are within the manuscript and its Supporting Information files.

## Abstract

Heme and iron metabolic pathways are highly intertwined, both compounds being essential for key biological processes, yet becoming toxic if overabundant. Their concentrations are exquisitely regulated, including via dedicated two-component systems (TCSs) that sense signals and regulate adaptive responses. HemKR is a TCS present in both saprophytic and pathogenic *Leptospira* species, involved in the control of heme metabolism. However, the molecular means by which HemKR is switched on/off in a signal-dependent way, are still unknown. Moreover, a comprehensive list of HemKR-regulated genes, potentially over-lapped with iron-responsive targets, is also missing. Using the saprophytic species *Leptospira biflexa* as a model, we now show that 5-aminolevulinic acid (ALA) triggers the shutdown of the HemKR pathway in live cells, and does so by stimulating the phosphatase activity of HemK towards phosphorylated HemR. Phospho~HemR dephosphorylation leads to differential expression of multiple genes, including of heme metabolism and transport systems. Besides the heme-biosynthetic genes *hemA* and the catabolic *hmuO*, which we had previously reported as phospho~HemR targets, we now extend the regulon identifying additional genes. Finally, we discover that HemR inactivation brings about an iron-deficit tolerant phenotype, synergistically with iron-responsive signaling systems. Future studies with pathogenic *Leptospira* will be able to confirm whether such tolerance to iron deprivation is conserved among *Leptospira* spp., in which case HemKR could play a vital role during infection where available iron is scarce. In sum, HemKR responds to abundance of porphyrin metabolites by shutting down and controlling heme homeostasis, while also contributing to integrate the regulation of heme and iron metabolism in the *L. biflexa* spirochete model.

**Funding:** AB was supported by grants #FCE_1_2017_1_136291 (Agencia Nacional de Investigacion e Innovacion https://www.anii.org.uy/); and #PIU_761 Pasteur International Joint Research Unit IMiZA (Institut Pasteur/Institut Pasteur de Montevideo https://www.pasteur.fr/en/ & https://pasteur.uy/en/) MP was supported by grant #PIU_761 Pasteur International Joint Research Unit IMiZA (Institut Pasteur/Institut Pasteur de Montevideo https://www.pasteur.fr/en/ & https://pasteur.uy/en/) JAI had a PhD fellowship POS_NAC_2016_1_ 129903 (Agencia Nacional de Investigacion e Innovacion https://www.anii.org.uy/) TC was supported by grants #ANR-10-INBS-09 (Agence Nationale de la Recherche https://anr.fr/en/) and IBISA (Infrastructures en Biologie, Sante et Agronomie https://www.ibisa.net/).

**Competing interests:** The authors have declared that no competing interests exist.

## Introduction

Bacteria detect intracellular and environmental stimuli through an array of sensory proteins that transmit the information, triggering physiologic adaptive responses. Such sensory/regulatory proteins constitute networks of signal transduction systems regulating numerous biologic processes [1]. Two-component systems (TCSs) are signal transduction systems [2, 3], most often composed of a transmembrane sensory histidine-kinase (HK) and a cytosolic response regulator (RR). The presence/absence of a specific stimulus regulates the activation of the HK through ATP-mediated histidine autophosphorylation [4]. The phosphorylated HK transfers its phosphoryl group to a conserved aspartate residue on its cognate RR [5], which can thereafter bind to DNA promoter sites, regulating the transcription of one or more genes. Most HKs are also capable to act as RR-specific phospho-aspartate phosphatases, critically contributing to keeping the pathways off when appropriate [4, 6, 7]. For such down-control function, some TCSs may also rely on dedicated auxiliary phosphatases, distinct from the sensory HK [8, 9].

The Spirochetes phylum includes unique Gram-negative bacteria with spiral-shaped cells and endoflagella confined within the periplasm [10]. *Borrelia burgdorferi* (the causative agent of Lyme disease) and *Treponema pallidum* (syphilis), lack the enzymatic machinery to synthesize heme [11], with *Borrelia* evolving to use manganese instead of iron altogether [12]. On the contrary, *Leptospira* spp. (including the etiologic agents of leptospirosis) need iron to survive [13, 14]. Able to import heme from the milieu, *Leptospira* also synthesizes it *de novo* via the C5 pathway (S1 Fig) [13, 15]. The metabolism of iron and heme are highly connected: heme is a tetrapyrrole that binds iron tightly within its porphyrin ring –an introduction that requires catalysis by a ferrochelatase–, allowing for heme's essential functions in electron transfer, diatomic gas sensing/transport, and catalysis of diverse biochemical reactions [16]. Heme oxygenases on the other hand, degrade heme in an oxygen-dependent manner, rendering biliverdin, carbon monoxide and free iron [17]. Heme synthesis and catabolism thus depend on–and modify–the concentrations of iron in the cell. Intracellular iron excess is highly toxic, unleashing the production of harmful oxygen radicals through Fenton reactions. Excess of heme has also been reported to be toxic [18], even though the mechanisms by which heme kills bacteria remain unclear.

In bacteria, TCSs have been reported to sense iron [19]. However, iron-signaling was shown to be mainly mediated via one-component systems, namely the ferric uptake regulator protein Fur and other iron-sensitive regulators such as DtxR/IdeR, RirA and Irr [20]. As for heme-responsive signaling, TCSs appear to play a dominant role in Gram positive bacteria [21, 22], whereas in Gram negatives, diverse examples point to extra-cytoplasmic function σ-factor signal transduction cascades, and direct/indirect sensing via iron-responsive Fur-like regulators [23]. In *Leptospira*, iron and heme are regulated in singular ways [13], with no canonical Fur factors, and a distinct TCS, HemKR, controlling heme metabolism [24]. Using *Leptospira biflexa*, a saprophytic model that is similar to pathogenic species yet easier to manipulate [25], we have previously shown that HemR is phosphorylated by HemK, and that phospho-HemR (P~HemR) binds to a *hem*-box (TGACA[N$_6$]TGACA motif) present upstream of a few heme-metabolism genes [26]. P~HemR was proposed to be a transcriptional activator of genes encoding for the heme biosynthetic pathway, and a repressor of *hmuO*, which encodes the heme-degrading enzyme heme oxygenase [26]. However, a more comprehensive examination of the HemR regulon was not available. Moreover, molecular signal(s) able to switch on/off the HemKR pathway, and the molecular mechanism by which such switching occurs in live cells, were still unknown.

In this work, we show that *L. biflexa* HemKR is shut down when the cells are exposed to extra-cellular 5-aminolevulinic acid (ALA), a committed porphyrin biosynthesis precursor.

The signaling mechanism occurs by specific stimulation of HemK's phosphatase activity over its cognate partner, P~HemR. HemR dephosphorylation then leads to the differential expression of multiple genes, including of heme metabolism and transport systems. Lastly, we demonstrate that, synergistically with other iron-responsive systems, HemKR is involved in *L. biflexa* adaptation to iron starvation, ultimately conducing to bacterial iron-depletion tolerance.

## Materials and methods

### Bacterial strains and growth conditions

For targeted mutagenesis of the *hemK/hemR* operon in *Leptospira biflexa*, a suicide vector containing a kanamycin resistance cassette flanked by 600 bp of the 5'-end of *hemK* and 3'-end of *hemR* was introduced in *L. biflexa* serovar Patoc strain Patoc 1 by electroporation with a Biorad Gene Pulser Xcell, as previously described [27]. Electroporated cells were plated on EMJH agar plates supplemented with 50 μg/ml kanamycin. Plates were incubated for 1 week at 30˚C and kanamycin-resistant colonies were screened for double homologous recombination events by PCR. The *hemK⁻/hemR⁻* double knockout mutant thus generated (named Δ*hemKR* in this study) was further confirmed by whole-genome sequencing. *Leptospira biflexa wt* and Δ*hemKR* strains were grown at 30˚C in Ellinghausen-McCullough-Johnson-Harris (EMJH) liquid media [28]. When needed strains' genotypes were confirmed by PCR with oligonucleotides HemK_Fw and HemR_Rev (S1 Table). For growth measurements, *L. biflexa wt* and Δ*hemKR* were grown at 30˚C in standard EMJH medium with no agitation. When cultures reached exponential phase ($OD_{420}$ ~0.25), a 1:100 inoculum of each strain was subcultured in EMJH medium without supplemented iron, in the presence of 0, 35, 70 or 175μM 2,2'-dipyridyl (DIP), at 30˚C with no agitation until reaching stationary phase (10 days). Bacteria motility and viability were examined using a dark field microscope (AxioImager A2.LED, Zeiss).

The shuttle expression plasmid pMaORI ($Spc^R$) [29] was used to express mutated HemK-encoding gene constructs in *Leptospira*. Cloning and pMaORI manipulation/preparation was done in *E. coli* strain π1 (Δ*thyA*) cells [30]. pMaORI was transformed into *E. coli* strain β2163 (Δ*dapA*) [30]. *E. coli* strains π1 and β2163 were grown in LB containing 50 μg/mL spectinomycin, the former strain supplemented with 0.3 mM thymidine, whereas the latter with 0.3 mM diaminopimelic acid (DAP) [30].

### Cloning of complementation plasmids

To complement *L. biflexa* Δ*hemKR* mutant, the wildtype *hemKR* allele harboring its own promoter region (349 bp upstream *hemK*'s start codon), was amplified from *wt L. biflexa* using primers HemR_pMaORI_Fw, HemR_pMaORI_Rev, HemK_pMaORI_Fw, HemK_pMaORI_Rev, p_hemKR_pMaORI_Fw, and p_hemKR_pMaORI_Rev (S1 Table). Site-directed mutagenesis was performed to generate a *hemK_{T102A}* mutant allele with primers HemK_{T102A}_Fw and HemK1_Rev (S1 Table) and inserted into pMaORI [29] by restriction-free cloning [31]. Correctly recombined pMaORI constructs were confirmed by BamHI/EcoRI digestion profile. Cloned pMaORI-*hemKR* and pMAORI-*hemK_{T102A}R* plasmids were then transformed into *E. coli* strains π1 and β2163 for downstream conjugation experiments.

### Conjugation

*E. coli* strain β2163 cells harboring the pMaORI-*hemKR* and pMaORI-*hemK_{T102A}R* plasmids were co-incubated with *L. biflexa* cells in EMJH liquid media supplemented with 0.3 mM DAP (when required), until $OD_{420}$ 0.3, according to standard procedures [32]. Conjugants were

incubated in solid EMJH agar plates supplemented with 50 μg/mL spectinomycin, at 30°C for 7–10 days. Selected colonies were grown in liquid EMJH medium supplemented with 50 μg/mL spectinomycin, and the presence of the plasmid was confirmed by PCR with primers HemK_Fw and HemK2_Rev (S1 Table).

## Expression and purification of recombinant proteins

pQE80L (Qiagen) plasmids harboring HemK$_{\Delta 80}$ (encoding a truncated version of the kinase, lacking the first 80 amino acids), HemK$_{\Delta 80,T102A}$ (HemK$_{\Delta 80}$ with threonine 102 substituted by alanine), HemR$_{REC}$ (receiver domain only) and HemR$_{FL}$ (full-length) ORFs, each with an encoded Tobacco Etch Virus (TEV) protease cleavage site for N-terminal His-tag removal, were transformed into *E. coli* Rosetta-gami(DE3)—for HemK—or TOP10F'—for HemR—cells. Transformed Rosetta-gami(DE3) cells were grown with agitation (250 rpm) in 2xYT medium supplemented with 100 μg/mL ampicillin at 37°C until OD$_{600}$ ~ 1. Induction was carried with 1 mM IPTG at 30°C for 6 h. Transformed TOP10F' cells were grown similarly, until OD$_{600}$ ~ 0.6. Induction was performed using 1 mM IPTG at 37°C for 3 h. Cells were harvested by centrifugation at 4000g, and pellets resuspended in 50 mM Tris pH 8, 500 mM NaCl, EDTA-free protease inhibitors (Roche) and 0.1 mg/mL DNase I (Sigma). Cells were disrupted with 1 mg/mL lysozyme followed by 1 cycle of freezing/thawing and high-pressure homogenization (Emulsiflex-C5, Avestin).

Soluble fractions were obtained by centrifugation at 30000g for 30 min, and supernatants subjected to Ni$^{2+}$ affinity chromatography (HisTrap, Cytiva) in buffer A (50 mM buffer Tris pH 8, 500 mM NaCl and 120 mM imidazole for recombinant HemK$_{\Delta 80}$ and HemK$_{\Delta 80,T102A}$ purification, or 20 mM imidazole for HemR$_{REC}$ and HemR$_{FL}$). Elution was performed with a gradient of buffer B (buffer A with added 750 mM imidazole for HemK$_{\Delta 80}$ and HemK$_{\Delta 80,T102A}$ purification, and 500 mM imidazole for HemR$_{REC}$ and HemR$_{FL}$). Eluted fractions were pooled and incubated overnight with TEV protease in dialysis buffer (50 mM Tris pH 8, 500 mM NaCl). Digested pools were subjected to a second Ni$^{2+}$ column purification step to separate cleaved His-tag and TEV protease from the sample. All samples were finally subjected to size-exclusion chromatography (Superdex 16/60 75 prep column, Cytiva), previously equilibrated with 25 mM Tris pH 8 and 500 mM NaCl. The peak fractions were pooled, concentrated, and stored at -80°C until use.

## Autophosphorylation, phosphoryl-transfer and dephosphorylation catalysis assays

For autophosphorylation assays, 20 μM HemK$_{\Delta 80}$ or HemK$_{\Delta 80,T102A}$ were incubated with 5 mM ATP, 10 mM MgCl$_2$. For phosphotransfer assays, 20 μM HemK$_{\Delta 80}$ or HemK$_{\Delta 80,T102A}$ previously incubated with 5 mM ATP, 10 mM MgCl$_2$ for 60 min at room temperature (RT), were then incubated with equimolar concentrations of HemR$_{REC}$ or HemR$_{FL}$. For phosphatase assays, 600 μM HemR$_{REC}$ or HemR$_{FL}$ were first phosphorylated by incubating with 5 mM acetyl phosphate, 10 mM MgCl$_2$, and then dimeric, phosphorylated HemR$_{REC}$ or HemR$_{FL}$, were purified from monomeric, non-phosphorylated forms, by size-exclusion chromatography (Superdex 10/300 75 prep, Cytiva). 20 μM of phosphorylated HemR$_{REC}$ or HemR$_{FL}$ were incubated with equimolar concentrations of HemK$_{\Delta 80}$ or HemK$_{\Delta 80,T102A}$. At defined time intervals, reactions were stopped with SDS buffer and 25 mM dithiothreitol (DTT) for 10 min at RT, followed by further incubation with 50 mM iodoacetamide for additional 10 min. All samples were then separated by PhosTag-SDS-PAGE, as described below. Gels were stained with Coomassie blue, scanned with a CanoScan Lide 110 (Canon) scanner, and bands quantified by densitometry using ImageJ [33].

## PhosTag-SDS-PAGE and Western blot of whole protein extracts

*L. biflexa* cultures were grown in 30 mL EMJH until $OD_{420}$ ~0.2. Prior to 5-aminolevulinic acid (ALA) or 2,2'-dipyridyl (DIP) incubation, an aliquot of 6.5 mL of untreated culture was harvested at 10000 g for 10 min at 4°C. When indicated, 300 μM ALA or 35 μM DIP was added to the cultures, and 6.5 ml aliquots were harvested at 1, 2 and 3 h post-incubation. Cell pellets were lysed resuspending with BugBuster and Lysonase according to the manufacturer's protocol (Merck). Lysed cells were fractionated by centrifugation at 16000 g for 20 min at 4°C, and an aliquot of the soluble fraction was resuspended in SDS buffer with 25 μM DTT and incubated 10 min at RT, followed by 50 μM iodoacetamide for extra 10 min at RT. Samples were run in SDS-PAGE 12% bis-acrylamide, co-polymerized with 100 μM PhosTag (Wako) and 200 μM $ZnCl_2$. PhosTag in the presence of transition metals, shifts the electrophoretic mobility of phosphorylated proteins [34, 35], allowing to quantify P~HemR/HemR ratios. Electrophoreses were run at <10°C. Before Western blotting, PhosTag was eliminated by washing gels with SDS-PAGE running buffer and 2mM EDTA for 10 min, and then with no EDTA for an extra 10 min. Gels were blotted to nitrocellulose HyBond (Cytiva) membrane overnight with a TE 22 Mighty Small Transphor Tank Transfer Unit (Amersham) at 20 V.

A polyclonal monospecific anti-HemR antibody (αHemR) was produced in rabbits. Immunizations were done at Instituto Polo Tecnológico de Pando (Uruguay) respecting ethical guidelines for the use of animals in research. Briefly, 100 μg pure recombinant HemR (produced in *E. coli*, see [26] for construct and purification procedures) were injected subcutaneously with complete Freund's adjuvant, followed by two boosters of 50 μg HemR in incomplete Freund's at 14 and 28 days post-first immunization. Animals were bled for intermediate titration assays, and at ~60 days post-immunization for sera separation, stored at -20°C until use. Western blot membranes were pre-blocked with 3% bovine serum albumin in 0.1% PBS-Tween for 3.5 h, at RT and gentle agitation. αHemR was diluted 1/100 in blocking solution, and incubated with the membrane for 2 h at RT, with gentle agitation. Membranes were washed four times with 0.1% PBS-Tween 15 min/each, with strong agitation. Secondary anti-rabbit IgG (Sigma) antibody diluted 1/20000 in blocking solution was added to the membrane, with gentle agitation for 1 h RT. Four washing steps were repeated as before, and membranes were finally incubated with bioluminescent reagents (BM Chemiluminescence Western Blotting Kit, Roche) for 5 min. Protein bands were revealed and quantified using an ImageQuant$^{TM}$ 800 (Cytiva) imaging system. Percentage of P~HemR (%P~HemR) was calculated by densitometry using ImageJ Fiji, subtracting background and subsequently quantifying total HemR signal ($HemR_{TOTAL}$), which is the sum of the signal corresponding to P~HemR plus the signal corresponding to non-phosphorylated HemR ($HemR_{NONP}$) ($HemR_{TOTAL}$ = P~HemR + $HemR_{NONP}$). Then, %P~HemR = P~HemR*100/$HemR_{TOTAL}$.

## RNA extraction

*L. biflexa wt* and Δ*hemKR* mutant strains were grown in liquid EMJH media with no shaking at 30°C until mid-exponential phase ($OD_{420}$ ~0.2). Untreated cultures and cultures treated with $FeSO_4$ 2 mM, δ-aminolevulinic acid (ALA) 300 μM, and 2,2'-dipyridyl (DIP) 35 μM were further incubated 3 h at 30°C with no shaking. RNA extraction was then performed as previously described [36].

## RNAseq

RNAseq libraries were prepared with 3 replicates for each condition, using a TruSeq Stranded mRNA library Preparation Kit (Illumina, USA) following the manufacturer's protocol. The libraries were sequenced on an Illumina NextSeq2000 platform, generating single-end reads

with an average read length of 107 bp. Quality control and data pre-processing were conducted to ensure robust downstream analyses. The number of reads was between 12-20M reads per sample, with an average phred score of 33. Transcription of the *hemK* and *hemR* genes were analyzed in detail in the Δ*hemKR* mutant strain as a direct control of the knockout. *hemR* (*LEPBIa1422*) transcription was indeed abolished in all conditions. As for *hemK* (*LEPBIa1423*), standard differential gene expression (DGE) analyses did not detect a significant shift with respect to the *wt*; yet, a more detailed mapping on the genomic loci revealed that the reads corresponding to *hemK*, comprise only the initial 5' fragment of the gene, upstream of the kanamycin-resistance cassette. Fold-change of individual mRNA species was an obvious figure to analyze DGE. The bioinformatic analyses of whole transcriptomes and differential gene expression were performed with Sequana [37]. Specifically, the RNAseq pipeline (v0.15.1) available at https://github.com/sequana/sequana_rnaseq was employed, built upon the Snakemake framework [38]. To prepare the data, reads were trimmed for adapter sequences and low-quality bases using fastp software v0.20.1 [39] and were subsequently mapped to the *Leptospira* genome using bowtie2 [40]. The nucleotide sequence and annotation were downloaded from the MicroScope platform [41] using *L. biflexa* serovar Patoc Patoc 1 entry (corresponds to NCBI genome RefSeq assembly GCF_000017685.1). The count matrix was generated using FeatureCounts 2.0.0 [42], which assigned reads to relevant features based on the previously mentioned annotation. To identify differentially regulated genes, statistical analysis was performed with the DESeq2 library 1.30.0 [43], and HTML reports were generated using the Sequana RNAseq pipeline. Key parameters for the statistical analysis encompassed significance, measured by Benjamini-Hochberg adjusted p-values with a false discovery rate (FDR) threshold of less than 0.05, as well as the effect size, quantified through fold-change calculations for each comparison. The *hem*-box DNA-binding motif was scanned with the FIMO Motif Search Tool (v. 5.5.1) [44].

## qRT-PCR

Transcription levels of target genes *hemA* (*LEPBIa1171*) and *hmuO* (*LEPBIa0669*) were evaluated in *L. biflexa wt* and Δ*hemKR* cells by quantitative real time PCR 9qRT-PCR). RNA extracted as described above, was used for cDNA synthesis using iScript cDNA Synthesis Kit (BioRad). PCR amplification was done on cDNAs using a QuantStudio 3 thermocycler (Applied Biosystems, Thermo Fisher Scientific), using the primers listed in supplementary S1 Table, and followed in real time by using SsoFast EvaGreen Super-mix (BioRad), as follows: 15 min at 95°C; 40 x (0.25 min at 95°C, 1 min at 60°C); 0.25 min at 95°C; melting curves 1 min at 60°C; 0.25 min at 95°C with increment 0.1°C/s. The relative expression of target genes was calculated using the delta-delta Cq method and normalized against *rpoB* mRNA levels as described before [26].

## Results

### The cytoplasmic catalytic region of HemK is trapped in the kinase-active state

A soluble construct of HemK (HemK$_{\Delta 80}$) lacking the first 80 amino acids that span the periplasmic and trans-membrane regions of the native protein, possesses ATP-dependent autokinase activity (Fig 1A), confirming previous observations [24]. Further characterizing the kinetics of the reaction, HemK$_{\Delta 80}$ achieved 50% maximal phosphorylation in ~60 min in the presence of excess ATP as phosphoryl donor (Fig 1A). P~HemK$_{\Delta 80}$ was also able to catalyze rapid phosphoryl-transfer to the phospho-receiver domain of HemR (HemR$_{REC}$) (Fig 1B),

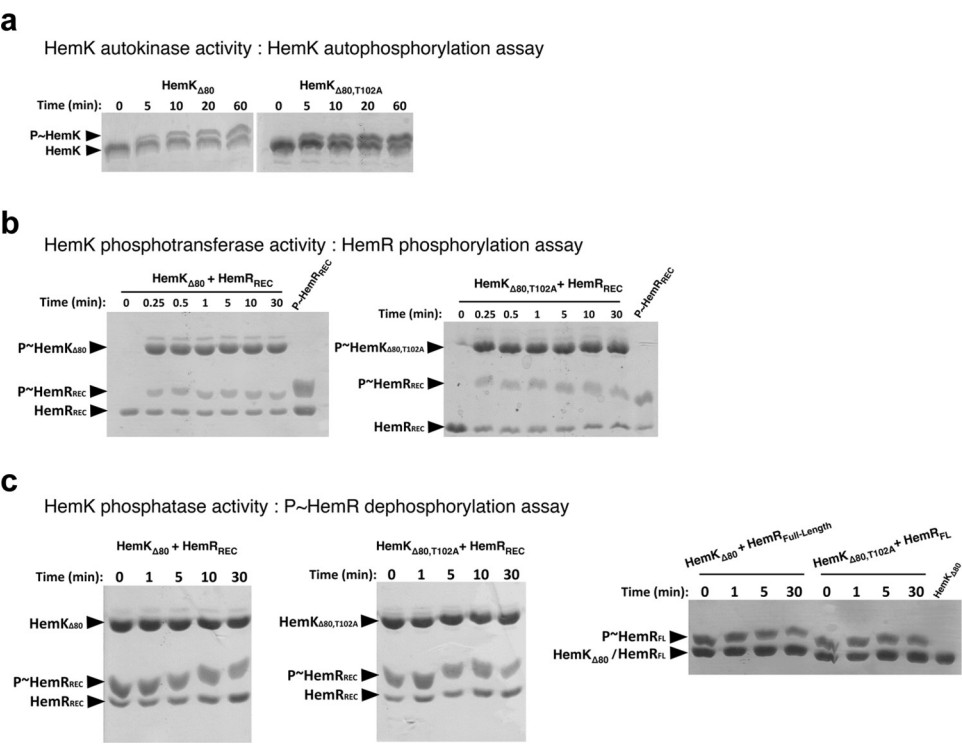

**Fig 1. Autokinase, phosphotransferase and phosphatase activities catalyzed by HemK. (a)** Autophosphorylation of HemK$_{\Delta80}$ (left panel) and the phosphatase-null point-mutant HemK$_{\Delta80,T102A}$ (right), incubated for indicated times with 5mM ATP-Mg$^{2+}$. **(b)** Phosphoryl-transfer kinetics from P~HemK$_{\Delta80}$ (left) or P~HemK$_{\Delta80,T102A}$ (right) to HemR$_{REC}$, in the presence of 5mM ATP-Mg$^{2+}$. **(c)** HemK-catalyzed phosphatase activity on P~HemR$_{REC}$, using HemK$_{\Delta80}$ (left panel) or HemK$_{\Delta80,T102A}$ (center). The dephosphorylation of the regulator was not detectable under any condition. 5mM ADP-Mg$^{2+}$ was included in the reactions, as some HisKA His-kinases are known to require nucleotide to exert their phosphatase activity (indistinguishable results were obtained with no nucleotide added). The right-most panel reproduces the previous panels' assays but using a full-length construct of HemR (HemR$_{FL}$), ruling out the potential need of its DNA-binding domain to detect HemK phosphatase activity. The phosphorylated species of HemK and HemR are indicated (P~HemK and P~HemR) and suffer a mobility shift during electrophoresis due to the co-polymerization of PhosTag into the polyacrylamide gels. Each experiment was performed in duplicate, representative Coomassie-stained gels are shown.

with undetectable back-transfer (Fig 1C), similarly to what occurs in other HK families [45]. Threonine 102 (T102) is predicted to play a key role in HemK's phosphatase activity, as this is a conserved residue in this HK family (HisKA), engaged in the dephosphorylation of the RRs' phospho-aspartate [4, 46]. Neither the autophosphorylation nor the phosphoryl-transfer reaction kinetics were affected by substituting HemK's T102 by an alanine (Fig 1A and 1B). Even though most HisKA HKs exhibit specific phosphatase activity towards their cognate P~RRs [4], HemK$_{\Delta80}$-dependent P~HemR dephosphorylation was not detected *in vitro* (Fig 1C). Different conditions were assayed, especially considering known requirements of other HisKA kinases to fully catalyze phosphatase reactions. Among these, the inclusion of ADP, and/or the use of full-length P~HemR as substrate (Fig 1C), did not result in HemK$_{\Delta80}$ phosphatase enhancement.

Together with previous evidence exploring the effect of the key phosphorylatable histidine (H98) and aspartate (D53) residues on HemK and HemR, respectively [24], we now show that HemK$_{\Delta80}$ specifically phosphorylates HemR in a unidirectional forward-transfer manner. Surprisingly, *in vitro* HemK$_{\Delta80}$ is not able to dephosphorylate P~HemR, altogether suggesting that HemK$_{\Delta80}$ is intrinsically trapped in a kinase-active state, in need of its sensory/

transmembrane portion to shift to a phosphatase-competent state under appropriate signaling conditions (as demonstrated below).

## The heme precursor 5-aminolevulinic acid triggers the stimulation of HemK phosphatase activity towards P~HemR

To discover signals sensed via the HemKR system, we asked whether the *L. biflexa* HemKR phosphorylation status was directly affected by exposition/deprivation of the cells to heme-synthesis building moieties (Fig 2). First attempts were done using heme itself. However, its toxicity at low micromolar concentrations, and the overwhelming global transcriptional effect we observed in *L. biflexa* cells exposed to <10µM hemin, hampered clearcut interpretations. Hence, a simpler scheme was followed using other heme-building blocks, such as 5-aminolevulinic acid (ALA) and iron. *L. biflexa* cells grown in EMJH medium until mid-exponential phase, were left untreated, or were supplemented with 300µM ALA, 35µM 2,2'-dipyridyl (DIP, a strong metal chelator) or 2mM $FeSO_4$ (Fig 2). Bacterial whole protein extracts were then separated with PhosTag-SDS-PAGE, and HemR was revealed by Western blot using an anti-HemR polyclonal antibody (Fig 2A). We observed that in normal growth conditions (untreated) *L. biflexa* P~HemR represents ~12–25% of the total pool of HemR in the cells, and that this ratio did not change throughout the course of the experiment (Fig 2B). Excess or deficit of iron also did not produce detectable changes of HemR phosphorylation. However, exposure to ALA provoked a sustained depletion of P~HemR, already observed at 1h of incubation (Fig 2A and 2B). To uncover the molecular mechanism of this ALA-triggered effect, we hypothesized that HemK directly or indirectly senses ALA (or a downstream metabolite derived from ALA, including fully synthesized porphyrins such as heme), leading to the

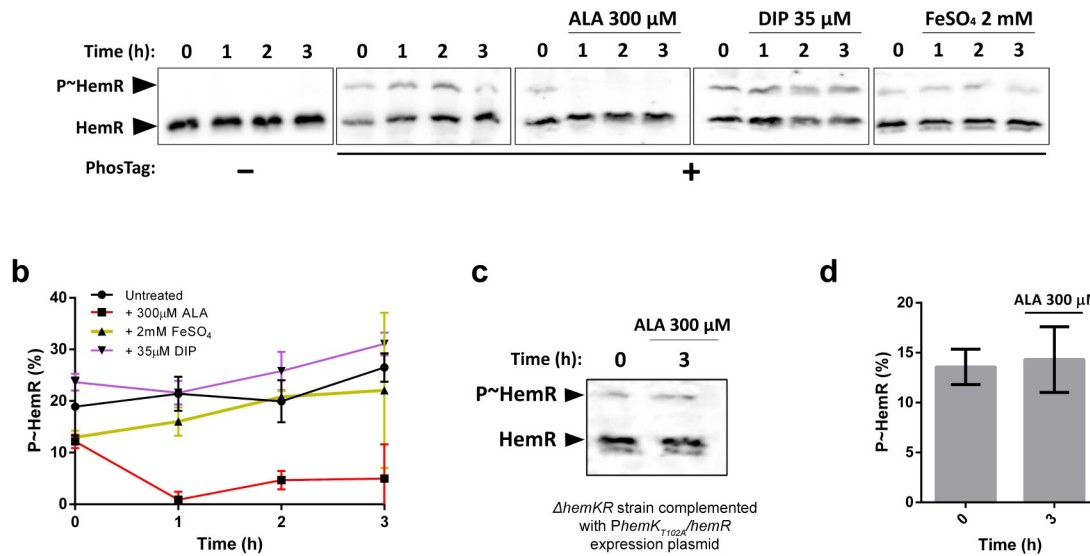

**Fig 2. 5-aminolevulinic acid, but not $Fe^{3+}$, shuts down HemK-mediated phosphorylation of HemR *in vivo*. (a)** PhosTag-SDS-PAGE of *wt L. biflexa* whole protein extracts, followed by Western blot using an anti-HemR antibody. Bacteria were left untreated, or treated with 5-aminolevulinic acid (ALA), 2,2-dipyridyl (DIP), or an excess of iron ($FeSO_4$), as indicated. Black arrows indicate the electrophoretic mobility position of phosphorylated and unphosphorylated HemR species. **(b)** Quantification of P~HemR levels from panel (a) shown as percentage. Measurements were obtained from three biological replicates. **(c)** PhosTag-SDS-PAGE followed by Western blot as in panel (a). *L. biflexa* Δ*hemKR* mutant strain complemented with plasmid P*hemK*$_{T102A}$/*hemR* was left untreated or treated with ALA for 3h. **(d)** Quantification of P~HemR levels from panel (c), shown as percentage. Measurements were obtained from three biological replicates.

activation of its phosphatase activity towards P~HemR. To test this hypothesis, we performed the same analysis using a *L. biflexa* Δ*hemKR* mutant strain [24] complemented with plasmid P*hemK*$_{T102A}$/*hemR*, encoding for HemK$_{T102A}$ phosphatase-null mutant and *wt* HemR [4, 46] under the control of the operon's native promoter. Notoriously, this strain exhibited comparable levels of HemR and P~HemR as the *wt* strain, yet did not trigger P~HemR dephosphorylation in response to ALA (Fig 2C and 2D).

Altogether, our results confirmed that HemK possesses phosphatase activity *in vivo*, leading to P~HemR dephosphorylation in a signal-dependent manner. Indeed, ALA, but not iron, was shown to be a potential signal to shut down the HemKR pathway.

## HemKR-dependent gene regulation is consistent with a response effecting heme/iron homeostasis

Previous efforts to define the HemR regulon had pinpointed two targets, which code for enzymes involved in heme biosynthesis and degradation [24, 26]. To further our understanding, we performed whole transcriptome analysis to identify HemKR-dependent differential gene expression (DGE) in *L. biflexa*, using ALA as an effective signal to shut down the pathway (S2 Table). We also used the wild-type strain (*wt*) and a double knockout with both *hemK* and *hemR* genes disrupted (Δ*hemKR*), to discriminate direct implications of HemKR in the ALA-triggered response (S2 Table). *L. biflexa* single gene knockout strains (Δ*hemK* and Δ*hemR*) had been previously characterized as auxotrophic for heme, a deficiency that could also be rescued by adding ALA to the growth medium [24]. Intriguingly, *L. biflexa* Δ*hemKR* strain grew in the absence of ALA/heme, even though no evident differences could be identified among single and double knockout strains' whole genome sequences, other than the targeted gene disruption sites. The growth of the Δ*hemKR* strain was also not affected by washing thoroughly and passing the cells eight consecutive times in EMJH medium without ALA/heme supplementation, altogether confirming that this double knockout mutant is capable of *de novo* heme biosynthesis.

Transcriptomic analyses confirmed that transcription of *hemK* and *hemR* genes was indeed abolished in the mutant strain (S2 Table). The kanamycin cassette introduced to disrupt the *hemK*/*hemR* operon locus, sits at base-pair 286 starting from the 5' end of *hemK*'s open reading frame. Hence, only short transcripts mapping onto this short 5' *hemK* fragment were detected, confirming that HemK protein cannot be expressed(see Experimental Procedures for further details). DGE analysis of ALA-treated *vs* untreated *wt Leptospira* cells, revealed a significant downregulation of 128 genes and upregulation of 51 (log$_2$ fold change (FC) > |1|). This contrasted with the effect in the Δ*hemKR* mutant, which showed only 6 downregulated and 24 upregulated genes under the same experimental conditions (Fig 3A), consistent with HemKR being engaged in mediating the ALA-triggered response. Many ALA-responsive genes encode hypothetical proteins, for which functional roles require future studies (S3 Table). Yet, several key heme and iron metabolism-related genes were found to be differentially expressed in the *wt* strain and not in the Δ*hemKR* mutant (Fig 3A and S3 Table). Among such targets, the heme biosynthesis operon *hemA* was down-regulated, as well as *exbB1/D1* and *LEPBIa3432*, the latter two linked to TonB-dependent iron/siderophore/porphyrin outer-membrane transport. On the other hand, the heme-degrading *hmuO* heme oxygenase, displayed modest yet significant overexpression. Two representative targets, up- and down-regulated, were further confirmed by qRT-PCR (Fig 3B and S4 Table).

That the dual log$_2$FC plot of differential expression results in a horizontally flat distribution (Fig 3A), uncovers the stronger transcriptional response of *L. biflexa wt* to ALA treatment in comparison to the Δ*hemKR* mutant strain. The behavior of the *hemA* and *exbB1* operons,

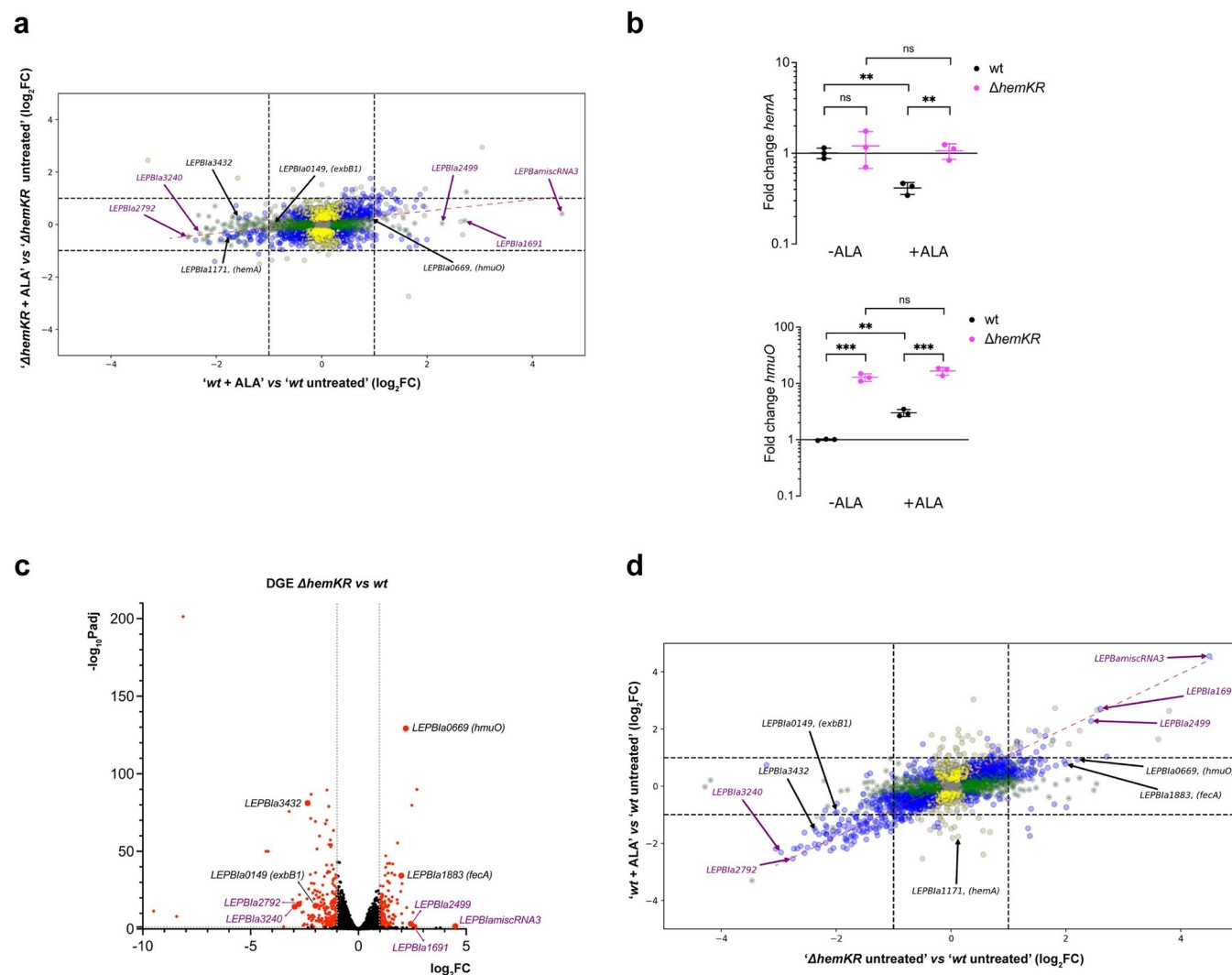

**Fig 3. 5-aminolevulinic acid induces a largely HemKR-dependent transcriptional regulation response. (a)** Transcriptomic differential gene expression (DGE) analysis of Δ*hemKR* and *wt L. biflexa* strains, comparing cells that were treated with 300μM 5-aminolevulinic acid (ALA) vs untreated (see S1 Table for full data). The plot is a dual log$_2$ fold change (log$_2$FC), comparing the DGEs of Δ*hemKR vs wt* strains. Note the stronger regulation in the *wt* strain compared to the Δ*hemKR* mutant (flat pattern of log$_2$FCs distribution; the red dashed line is the linear least squares regression curve with slope 0.1954 ± 0.011, and coefficient of determination R$^2$ = 0.084). Blue dots represent genes with P-values (Padj)≤0.05 in both strains; green dots, genes with Padj≤0.05 only in L. biflexa wt; yellow dots, genes with Padj≤0.05 only in *L. biflexa* Δ*hemKR*; grey dots, genes with Padj>0.05. Black arrows label selected genes with functional annotation or clearcut homology to functionally characterized orthologs involved in heme/iron metabolism; purple arrows, correspond to hypothetical genes or non-protein-coding RNAs. **(b)** Confirmation of DGE effects triggered by ALA, performing qRT-PCR of two target genes, one repressed by ALA (*hemA*) and one induced (*hmuO*). Statistical significance of differences was calculated by Student's t test using three replicas (ns = P>0.05; * = P≤0.05; ** = P≤0.01;*** = P≤0.001). **(c)** Volcano representation of significance *vs* fold change analyzing the transcriptomic DGE between *L. biflexa* Δ*hemKR vs wt* strains grown under standard EMJH culture conditions. Labeled genes are the same as in panel (a), except for *hemA* (here not significant with log$_2$FC -0.4191). **(d)** Dual log$_2$FC plot comparing '*wt* ± ALA treatment' vs 'Δ*hemKR vs wt*'. Note the diagonal pattern of distribution (least squares regression in red dashed line, slope 0.4732 ± 0.009, and coefficient of determination R$^2$ = 0.301) revealing a significant proportion of coincident differentially regulated genes in *wt*+ALA compared to Δ*hemKR*.

expected to be activated by P~HemR, was consistent with ALA triggering HemKR shutdown. However, fold-change figures were modest. To obtain further confirmatory evidence, we then asked whether the removal of HemR, in the Δ*hemKR* strain, could be a relevant proxy of HemR dephosphorylation. Transcriptomic DGE was performed, comparing *L. biflexa* Δ*hemKR vs wt* strains grown in standard EMJH conditions (Fig 3C and S2 Table). The elimi-nation of HemKR resulted in overall similar effects to those visualized with the *wt* strain under

ALA exposure. Moreover, the Δ*hemKR* knockout exhibited even higher fold-change figures (Fig 3A and 3C). The dual log2FC plot in Fig 3D shows an overall linear relationship in the direction of differential expression suggesting a similar transcriptomic effect of the exposure to ALA and *hemKR* deletion, both leading to HemKR shutdown. The *hemA* operon was a notable exception, repressed in the *wt* after exposure to ALA while not significantly affected in the Δ*hemKR* mutant, it strongly suggests a composite control by different regulatory elements, which has been reported for this gene cluster in other bacteria [47].

Several of the differentially regulated genes, both in *wt*+ALA and in the Δ*hemKR* mutant, comprise *hem*-box motifs within their promoter regions, such as *hmuO* and *hemA*, the expression of which is indeed known to respond to HemR control [26]. We now demonstrate that *exbB1/exbD1* and *LEPBIa3432* are also part of the HemR regulon. While the former had been previously anticipated [26], we have now identified a new motif **TGACA**GTACTG**TGACA** on the minus strand of the genome (nucleotide coordinates 3,550,754–3,550,769) [44], compatible with the *hem*-box consensus, and just upstream of *LEPBIa3430*/*LEPBIa3431*. These two genes were indeed downregulated, similarly to *LEPBIa3432*, altogether suggesting they may form an operon. Several other ALA-responsive targets do not display easily discernible *hem*-box motifs, yet the majority correspond to genes encoding hypothetical proteins, justifying focused studies to unveil their regulatory mechanism and functional assignments.

In sum, ALA–and/or some derived metabolite along the porphyrin biosynthesis pathway– shuts down the HemKR system, anticipating heme synthesis inhibition (due to HemA decrease) and simultaneous stimulation of heme degradation (due to HmuO heme oxygenase's surge).

## The phenotype of the ΔhemKR knockout strain revealed marked tolerance to iron deficit

Notoriously, *LEPBIa1883* lies among the most significantly upregulated genes in the *L. biflexa* Δ*hemKR* strain compared to *wt* (Fig 3C and 3D). This gene codes for the outer-membrane TonB-dependent ferric citrate receptor/transporter FecA. The expected effect would be for a larger pool of iron to become available in this mutant that lacks the HemKR system, a convergent effect with the one induced by heme oxygenase, which also liberates iron from heme. We thus conjectured that the Δ*hemKR* strain would be more tolerant to environmental iron deficit than the *wt*. To test this hypothesis, we assessed cell growth under increasing amounts of the iron-chelator DIP, producing mild to severe iron deficiencies. While the *wt* strain displayed the expected sensitivity to iron deprivation, *L. biflexa* Δ*hemKR* was significantly more resistant (Fig 4). Complete depletion of iron can be achieved by further increasing the concentration of DIP (175 μM), at which point both *wt* and Δ*hemKR* cultures arrested growth, only to be reverted–in both strains–by adding the xenosiderophore deferoxamine. All evidence considered, we confirmed that the *L. biflexa* Δ*hemKR* mutant, lacking the HemKR TCS, is indeed more tolerant to iron deficiency than the *wt* strain.

## Iron-deficiency triggers a transcriptional response that implicates a complex regulatory network beyond HemKR

The tolerance of *L. biflexa* Δ*hemKR* to iron deprivation suggested a crosstalk between heme- and iron-responsive regulatory pathways. We studied the systemic effect of iron on gene expression, in the *wt* and Δ*hemKR* strains. Iron-starving conditions were brought about by adding 35μM DIP to the culture, whereas iron excess was attempted by using 2mM $FeSO_4$-

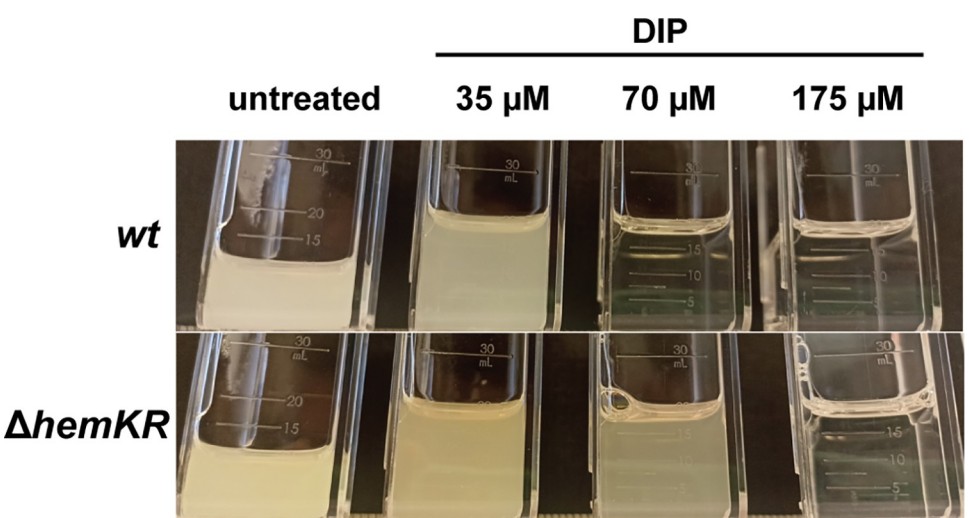

**Fig 4. *L. biflexa ΔhemKR* strain is more tolerant to iron deficiency.** *L. biflexa wt* and *ΔhemKR* were grown in EMJH medium with no supplemented iron. Cultures were left untreated or treated with increasing concentrations of the iron chelator 2,2'-dipyridyl (DIP) and incubated at 30˚C with no agitation for 10 days. One representative experiment is shown from three biological replicates.

supplemented medium (instead of 330 µM in standard EMJH). Unfortunately, iron overabundance hindered RNA-purification in the *wt* strain and could not be further analyzed (S2 Fig).

RNAseq analysis of *L. biflexa* under iron-restricted conditions revealed a strong regulatory response in both *wt* and *ΔhemKR* mutant (Fig 5A and S2 Table). Largely consistent with reports studying *L. interrogans* [14] and other bacteria [48] under similar stress, *L. biflexa wt* cells responded to $Fe^{3+}$ deficit by downregulating 151 genes and upregulating 105 (considering DGEs of $log_2FC > |1|$).

Although many of the iron-responsive genes code for hypothetical proteins, annotated genes linked to heme and/or iron metabolism were significantly affected (S5 Table), and two representative targets were further confirmed by qRT-PCR (Fig 5B and S4 Table). Even though the pattern of the iron-deficit response was globally similar in *wt vs ΔhemKR* (upper right and lower left areas along the diagonal in Fig 5A), a significant number of genes displayed fold-change differences between both strains, strongly suggesting a synergistic overlap of HemKR and additional iron-responsive regulatory systems. For instance, a few genes encoding hypothetical proteins were over-expressed (*e.g. LEPBIa1691, LEPBIa0249*, among ~25 others), whereas most were downshifted (*LEPBIa2210, LEPBIa2180*, among ~100 others) only in the *wt* strain, and not in *ΔhemKR* (Fig 5A). Also of note, 17 tRNA-encoding genes (spanning tRNAs that bind to 13 different amino acids) were intriguingly overexpressed, only when the HemKR system is present. Importantly, the absence of the HemKR system, abolished the wild-type iron-sensitive behavior of two operons that play key functions in heme homeostasis (*hemACBLENG* and *exbB1/exbD1*), which are repressed in the *wt* strain under iron-deficit (Fig 5A). Instead, in the *ΔhemKR* mutant *exbB1/exbD1* was no longer repressed and, more dramatically, the heme-biosynthesis operon *hemACBLENG* was induced ~2.5-fold. This means that the *ΔhemKR* strain exhibited ~8-fold increase of *hemACBLENG* expression compared to the *wt* under iron deprivation (S2 Table), strikingly indicating a higher heme biosynthetic activity when the metal is restricted, that requires the HemKR system to be shut down.

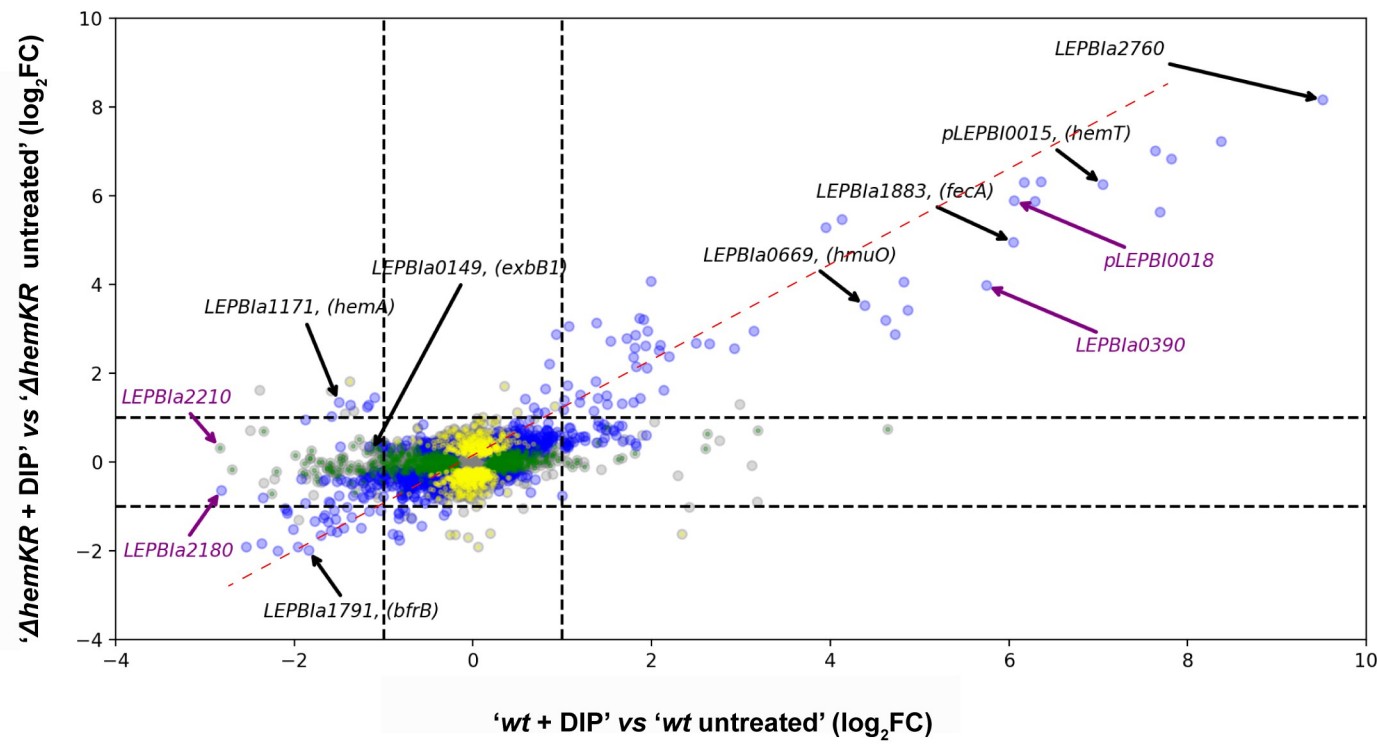

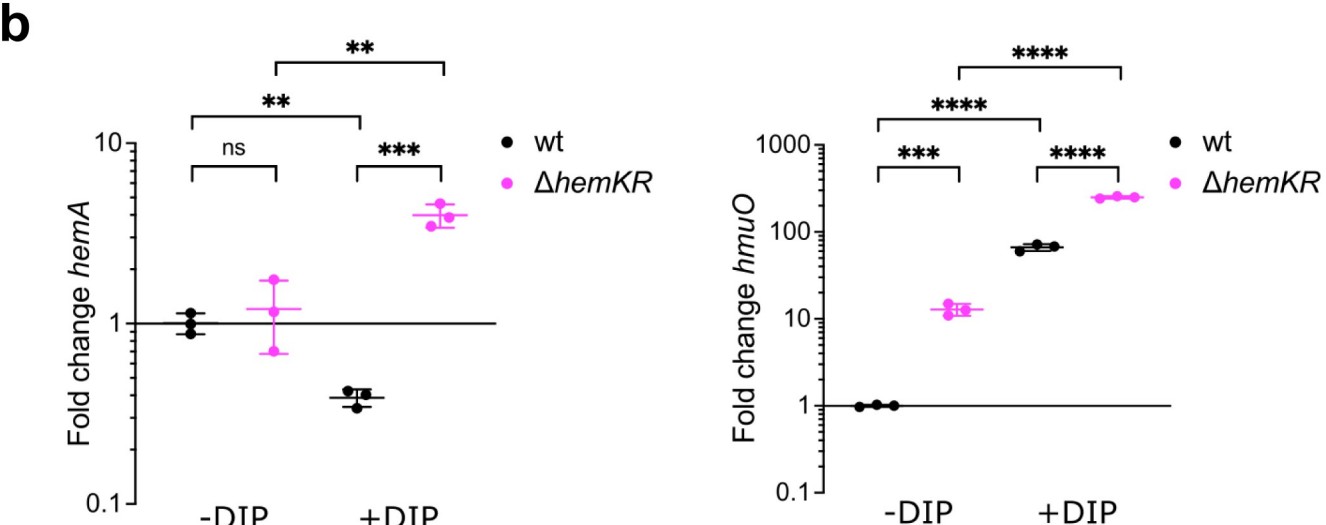

**Fig 5. Iron deficit induces transcriptional regulation with mixed HemKR-dependent/independent responses.** Transcriptomic differential gene expression analysis comparing *L. biflexa* cells treated *vs* untreated with 35 μM DIP for 3 h at 30°C with no agitation. **(a)** Dual log₂ fold change (log$_2$FC) plot, comparing differential gene expression (treated with DIP *vs* untreated) of Δ*hemKR vs wt* strains. Note the same trend in regulation behavior between *wt* and mutant strains (diagonal pattern of log$_2$FCs distribution; the red dashed line is the linear least squares regression curve with slope 0.5349 ± 0.009, and coefficient of determination $R^2$ = 0.456). Coloring and labeling schemes as in Fig 3A. **(b)** Confirmation of DGE effects triggered by DIP, performing qRT-PCR of the same target genes as in Fig 3B (*hemA*, *hmuO*). Statistical significance of differences was calculated by Student's t test using three replicas (ns = P>0.05; * = P≤0.05; ** = P≤0.01;*** = P≤0.001; **** = P≤0.0001).

All the evidence considered, a synergistic effect between HemKR and some other regulatory systems, facilitate an adaptive heme-homeostatic response when iron is scarce.

## Discussion

*Leptospira* cells require iron to live and are able to adapt in response to the metal's availability [13, 14]. *Leptospira* can also synthetize heme *de novo* [15], a study that later led to the identification [24] and characterization [26] of the two-component system HemKR as a regulator of heme metabolism. Besides its role as an essential cofactor of proteins involved in respiration and enzymatic catalysis, heme is well known as a diatomic gas sensor/transporter [16]. An $O_2$-sensing pathway in *Leptospira interrogans* is mediated by heme moieties bound to the three intracellular Per-ARNT-Sim (PAS) domains of *Li*Aer2, a trans-membrane methyl-accepting chemotaxis protein (MCP) [49]. An ortholog of *Li*Aer2 is present in *L. biflexa* (encoded by gene *LEPBIa1582*), conserving one of the cytoplasmic PAS domains, and suggesting that a similar gas-sensing role could be operational in the saprophytes. The direct connection between $O_2$ and reactive oxygen species, also links heme and iron metabolism to oxidative stress response in bacteria [20]. Indeed, the transcription of *hemK* and *hemR* is up-regulated when *L. interrogans* is exposed to sublethal $H_2O_2$ concentrations in a PerR-independent manner [50]: an activated HemKR pathway would be consistent with a tendency to reduce the pool of reactive iron in the cell, a common theme in bacterial adaptation to oxidative stress [51]. However, the HemKR-encoding genes are located in distinct genomic loci comparing pathogenic vs saprophytic *Leptospira* species [24].

Two target genes, *hemA* and *hmuO*, were known to be respectively activated and repressed by P~HemR in *L. biflexa* [26], with additional genes suspected to be part of the regulon. Relevant questions remained, not only concerning the composition of the full regulon, but also regarding the signals and molecular mechanisms that control the phosphorylation status of the HemKR pathway in live cells. We now report that 5-aminolevulinic acid (ALA) triggers the phosphatase activity of the sensory kinase HemK in live *L. biflexa* cells, promoting the dephosphorylation of its cognate response regulator HemR (Fig 6).

The cytoplasmic, catalytically competent region of HemK, appears to be trapped in a constitutive kinase-on state, requiring the sensory/trans-membrane region of the protein to switch to a phosphatase configuration under the right signaling. This signal-driven dephosphorylation depends specifically on HemK's phosphatase activity, eventually shutting down the HemKR pathway. Four residues away from the phosphorylatable histidine, HisKA family HKs (such as HemK) harbor an extremely conserved residue—a threonine or an asparagine—, critical to catalyzing the hydrolytic dephosphorylation of their cognate P~RR partner [4, 46]. That position is occupied by threonine 102 in HemK, which after substitution by an alanine (T102A), proved that the phosphotransferase activity of $HemK_{T102A}$ was not affected, yet its phosphatase activity was abolished, and with it, the capacity of $HemK_{T102A}$ to sense and respond to ALA (Fig 2). Cues that turn off TCSs are well known, and play important biological roles such as in quorum sensing [52], chemotaxis [53] and virulence regulation [54], among many others [7].

ALA is likely imported [55] in *L. biflexa* cells, and metabolized into heme and other porphyrins (S1 Fig). Under the conditions of the experiments, we cannot yet distinguish whether ALA and/or any of the intermediary compounds along the porphyrin biosynthesis pathway, including heme itself, might be acting as specific signals for HemK (Fig 6). Heme's toxicity on live *Leptospira* cells, and the extremely pleiotropic transcriptional response it triggers at sublethal concentrations, became challenging hurdles to address this question unequivocally. Nevertheless, ALA is a committed precursor of heme biosynthesis and thus well suited to act as a

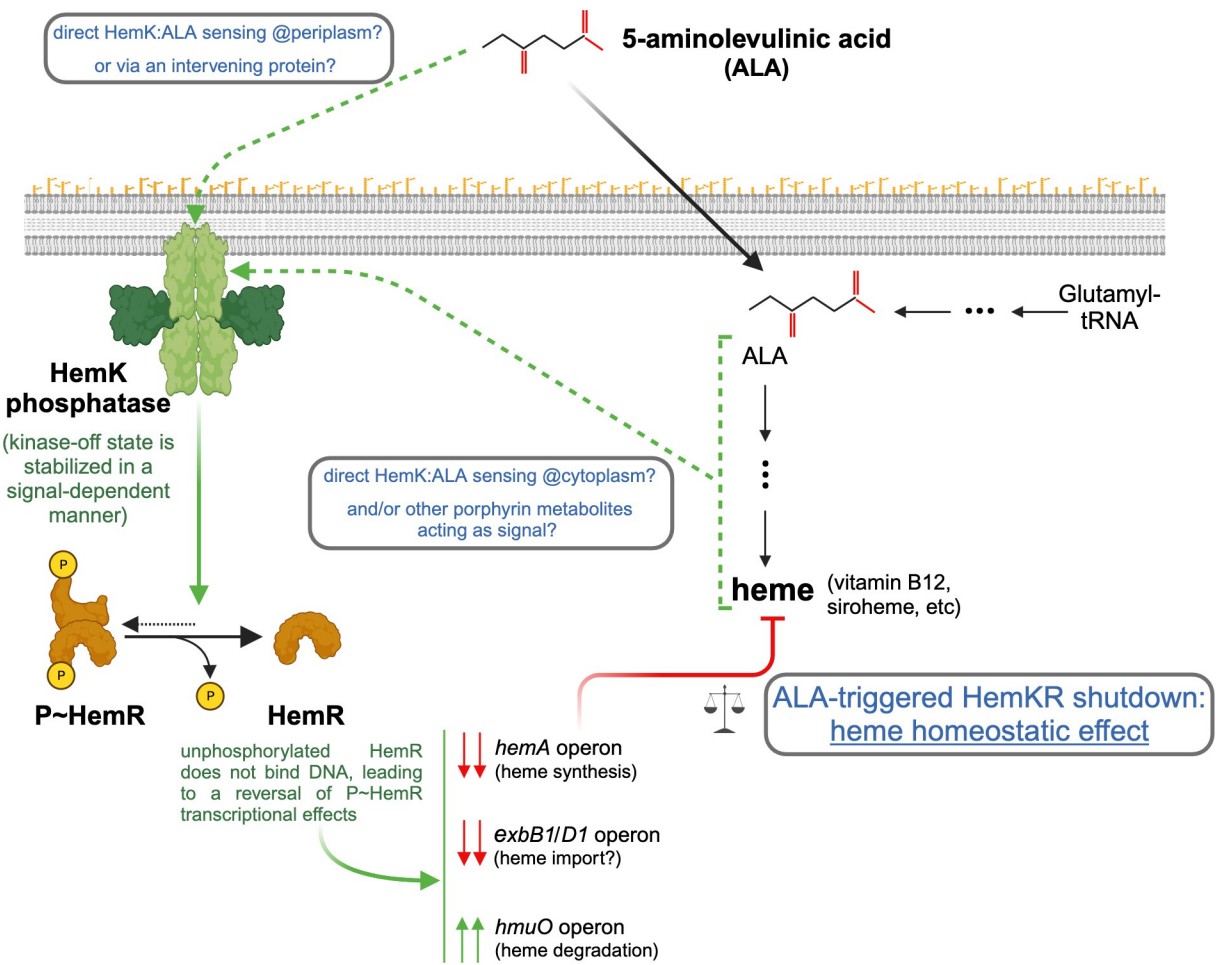

**Fig 6. Mechanistic hypothesis of ALA-triggered HemKR shutdown.** Signal-dependent stabilization of HemK in its kinase-off/phosphatase-on state, induces P~HemR dephosphorylation. Consequently, the P~HemR-dependent transcription of several heme-related genes/operons, is regulated (green double-arrows: up-regulation; red double-arrows: down-regulation), ultimately conducing to heme homeostasis. Further studies will reveal whether extracellular ALA is the cognate signal for HemK, and/or other metabolites along the porphyrin synthesis pathways. Green arrows mark inducing effects, whereas red stands for inhibition. ALA is likely transported from the extracellular milieu (black thick arrow), although HemK does not possess typical sensory domains on its intra-cytoplasmic portion. Thin black arrows depict metabolic routes to synthetize ALA and heme/porphyrins (intervening dots highlight several steps within). Created with BioRender.com.

proxy of the cell's heme-biosynthesis potential. Future biophysical/biochemical studies, using full-length trans-membrane HemK (*e.g.* with reconstituted proteoliposomes), are likely options to assess direct binding of candidate cues and subsequent kinase/phosphatase activity modulations. ALA is secreted by some bacteria [56], and secretion exacerbated when heme biosynthesis is disrupted [57]. Added to the fact that extracellular ALA can play strong signaling roles in Biology [58], it will be relevant to test in future investigations whether leptospires can secrete ALA. If so, especially under high cell density conditions (*e.g.* biofilms [59]), heme/iron homeostasis could be regulated at the population level via HemKR signaling.

Our transcriptomics approach to define a HemR regulon confirmed the two targets known to be regulated by phosphorylated HemR (P~HemR), namely the *hemACBLENG* gene cluster and *hmuO* that encodes heme oxygenase. As expected from previous reports [26], *hemA* and *hmuO* were found to be down- and up-regulated respectively after exposure to ALA. In that study, Morero *et al.* had predicted that the transcription of the *exbB1/exbD1* gene cluster

might also be controlled by P~HemR, given the presence of a HemR-binding *hem*-box within its promoter region. Indeed, we now report that *exbB1/exbD1* is significantly repressed when the HemKR TCS is shut down by ALA (Fig 3A). ExbB1/ExbD1 is an inner-membrane subcomplex that acts in concert with TonB, to energize outer-membrane transporters engaged in iron/siderophore/heme import [60]. The shutdown of HemKR uncovered additional target genes of the regulon, such as *LEPBIa3432*, which has not been annotated, yet codes for a protein displaying clear homology to PhuR-like, TonB-dependent outer-membrane receptor/transporters [61]. Together with ExbB1/ExbD1, a decrease of a putative heme receptor such as LEPBIa3432, concurrent with diminished HemA-dependent biosynthesis, anticipates a convergent down-control of heme concentrations (Fig 6). Added to heme oxygenase upregulation, when ALA is high (*i.e.* high heme-biosynthesis potential) cells maintain heme homeostasis which, if uncontrolled, would result in porphyrin overabundance.

Confirmation of the global expression effects of ALA-triggered HemKR shutdown, was obtained using the *L. biflexa* Δ*hemKR* mutant strain, which lacks HemR altogether. This mutant, a suitable proxy of HemR dephosphorylation, uncovered additional effects linking heme metabolism to iron availability. The overexpression of the iron transporter FecA, led to the hypothesis that the Δ*hemKR* mutant could be more tolerant to iron deficit. Indeed, a higher availability of intracellular iron due to increased transport (FecA) and augmented heme degradation (HmuO), could explain the greater tolerance of this mutant to iron deficit compared to *L. biflexa wt*. This scenario is also consistent with the strong overexpression of *LEPBIa1517*, which encodes a protein reliably predicted [62] as a class 2 polyphosphate kinase PPK2 [63]. PPK2s act as iron chelators, protecting bacterial cells from Fenton reaction toxicity [64] linked to higher iron concentrations.

Our analysis of the transcriptomic response to iron depletion was largely consistent with two previous studies on saprophytic and pathogenic *Leptospira* species [14, 65]. Yet, we now uncovered a fairly large number of iron-responsive genes that were only affected in the *wt* strain and not in the Δ*hemKR* mutant (~25 upregulated and >100 down-shifted), suggesting synergistic effects of HemKR and other signaling systems. Iron deprivation triggers a very strong regulatory response maximizing the entry of iron via heme and iron transporters, as well as mobilizing the intracellular iron pool by inducing heme demetallation–via heme oxygenase HmuO–, while repressing bacterioferritin-sequestered iron storage and heme biosynthesis (Fig 5). A direct role of the HemKR TCS is evidenced by the fact that the Δ*hemKR* mutant strain displays an altered expression of target genes, and more strikingly, inverts the effect on the heme-biosynthetic *hemA* operon. Additional transcription factors are yet to be identified to explain the overexpression of the *hemACBLENG* gene cluster in Δ*hemKR* when iron is lacking. The upregulation of both catabolic and biosynthetic heme enzymes in the DIP-treated Δ*hemKR* strain might be seen as an intriguing paradox. It is important to highlight that heme must be kept controlled when the heme-synthesis potential is high (such as under porphyrin precursors abundance). However, upon a deficit of environmental iron, cells must reallocate metabolic resources to release intracellular iron, while also securing enough heme to survive. It is tempting to speculate that this kind of reprogramming could account for the iron-deficit tolerance phenotype of *L. biflexa* Δ*hemKR* cells. Further investigations shall test this hypothesis, which predicts that HemKR's ability to respond to ALA offers *Leptospira* cells a selective advantage under iron-limiting conditions, a typical scenario for pathogenic species during host infection [66].

## Supporting information

**S1 Fig. Heme biosynthesis in *Leptospira*.** As in many other bacteria, *Leptospira* spp. possess all the genes coding for anabolic enzymes of the so called C5 pathway for de novo heme

biosynthesis. Starting from glutamyl-tRNAGlu, as initial precursor, the committed substrate 5- aminolevulinic acid (ALA) is generated (boxed in red). Four ALA moieties are needed to build the tetrapyrrole backbone of porphyrins including heme. Note that HemC and HemD are found within a single bifunctional enzyme in *Leptospira* spp [15]. The first step of heme degradation, catalyzed by heme oxygenase, is highlighted at the end of the pathway.
(TIF)

**S2 Fig. Environmental iron excess hindered RNA-purification from *wt L. biflexa*, and not from the Δ*hemKR* KO mutant.** Samples, as labeled, were submitted to standard RNA extraction procedures in preparation for whole mRNA transcriptomic sequencing. For unknown reasons, the wt strain–and not the Δ*hemKR* KO–exhibited an abnormal behavior if previously treated with excess iron: an insoluble yellow precipitate was formed during RNA extraction, which impaired RNA recovery prior to sequencing.
(TIF)

**S1 Table. List of oligonucleotides.** Used in this study for genotyping, mutagenesis and qRT-PCR, as indicated.
(XLSX)

**S2 Table. Normalized transcriptomic read counts.** The expression of each gene after normalization (see Materials and methods) is tabulated as read counts. Untreated cultures of wild-type *L. biflexa*, are compared to cultures treated with 5-aminolevulinic acid (ALA) or with the iron-chelator 2,2'-dipyridyl (DIP). Independent replicas are labeled with indices 1, 2 and 3 on the columns' names. Note there are 9 different sheets in the Excel table: the first two (Sheets 1 and 2) correspond to normalized read counts for the *wt* and the Δ*hemKR* strains respectively, with column names marking whether they come from untreated samples or following treatment with ALA or DIP. Sheets 3 and 4: Differential Gene Expression (DGE) analyses comparing ALA *vs* untreated conditions, respectively for the *wt* and Δ*hemKR* mutant strains. Sheets 5 and 6: DGE analyses comparing DIP *vs* untreated conditions, respectively for the *wt* and Δ*hemKR* mutant strains. Sheets 7–9: DGE analyses comparing expression of Δ*hemKR* mutant *vs wt* strains, respectively with no treatment, treated with DIP, and treated with ALA.
(XLSX)

**S3 Table. Selected genes differentially expressed when *wt L. biflexa* cells are exposed to 5-aminolevulinic acid.**
(DOCX)

**S4 Table. Quantitative reverse transcriptase-PCR of selected genes.** Relative amounts of mRNA of each gene are expressed with respect to untreated *wt* cells. Total mRNA extracted from *L. biflexa* cells (*wt* or Δ*hemKR* KO mutant strains as labeled), under identical treatments as the ones followed for the RNASeq experiments. Figures are averages from n = 3 replicas, with standard deviations (SD).
(XLSX)

**S5 Table. Selected genes differentially expressed when *wt L. biflexa* cells are exposed to 2,2'-dipyridyl (iron deficit).**
(DOCX)

**S1 Raw images. Uncropped original blot and gel images.**
(PDF)

## Acknowledgments

We thank Horacio Botti for initial preliminary experiments and fruitful discussions.

## Author Contributions

**Conceptualization:** Juan Andrés Imelio, Thomas Cokelaer, Alejandro Buschiazzo.

**Data curation:** Thomas Cokelaer.

**Formal analysis:** Felipe Trajtenberg, Sonia Mondino, Mathieu Picardeau, Alejandro Buschiazzo.

**Funding acquisition:** Thomas Cokelaer, Mathieu Picardeau, Alejandro Buschiazzo.

**Investigation:** Juan Andrés Imelio, Felipe Trajtenberg, Sonia Mondino, Leticia Zarantonelli, Iakov Vitrenko, Laure Lemée, Thomas Cokelaer.

**Methodology:** Juan Andrés Imelio, Iakov Vitrenko, Laure Lemée, Thomas Cokelaer.

**Project administration:** Alejandro Buschiazzo.

**Resources:** Juan Andrés Imelio, Felipe Trajtenberg, Leticia Zarantonelli, Mathieu Picardeau, Alejandro Buschiazzo.

**Supervision:** Alejandro Buschiazzo.

**Validation:** Leticia Zarantonelli, Thomas Cokelaer, Mathieu Picardeau.

**Writing – original draft:** Juan Andrés Imelio, Felipe Trajtenberg, Mathieu Picardeau, Alejandro Buschiazzo.

**Writing – review & editing:** Juan Andrés Imelio, Felipe Trajtenberg, Sonia Mondino, Leticia Zarantonelli, Mathieu Picardeau, Alejandro Buschiazzo.

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
