## [Decision Letter · Decision Letter 0]

14 Aug 2024

PONE-D-24-28906Signal-sensing triggers the shutdown of HemKR, regulating heme and iron metabolism in the spirochete Leptospira biflexaPLOS ONE

Dear Dr. Buschiazzo,

Thank you for submitting your manuscript to PLOS ONE. After careful consideration, we feel that it has merit but does not fully meet PLOS ONE’s publication criteria as it currently stands. Therefore, we invite you to submit a revised version of the manuscript that addresses the points raised during the review process. I agree with the second reviewer's suggestion that these results should primarily focus on the role of this system in L. biflexa. There may be similarities with pathogenic Leptospira spp, especially with those bacteria's aquatic environmental stages, but any implications on mammalian infection are only hypothetical. It will not be necessary to perform any of these studies with a pathogenic Leptospira species. L. biflexa and L. interrogans are distinct organisms.

We look forward to receiving your revised manuscript.

Kind regards,

Brian Stevenson, Ph.D.

Academic Editor

PLOS ONE

Journal Requirements:

Reviewers' comments:

Reviewer's Responses to Questions

**Comments to the Author**

1. Is the manuscript technically sound, and do the data support the conclusions?

Reviewer #1: Partly

Reviewer #2: Yes

2. Has the statistical analysis been performed appropriately and rigorously? 

Reviewer #1: Yes

Reviewer #2: Yes

3. Have the authors made all data underlying the findings in their manuscript fully available?

Reviewer #1: Yes

Reviewer #2: No

4. Is the manuscript presented in an intelligible fashion and written in standard English?

Reviewer #1: Yes

Reviewer #2: Yes

5. Review Comments to the Author

Reviewer #1: Comments on the manuscript_ PONE-D-24-28906_reviewer:

This is a well written manuscript. They investigated that how environmental signal sensing triggers the shutdown of HemKR, a two-component system (TCS) and regulates the homeostasis of Iron and heme metabolism in a saprophytic spirochete Leptospira biflexa.

The study is well designed, well executed and moreover results are well discussed. But the study is limited to a non-pathogenic leptospiral strain which lack the virulence factors associated with pathogenic Leptospira sp. Since the study aimed at exploring the influence of iron/heme metabolism and their response receptors (RR) and TCS target genes essential in regulating the heme homeostasis of Leptospira during infection. The authors claimed 5-aminolevulinic acid (ALA), a porphyrin precursor triggers the shutdown of HemKR pathway by inducing the HemK phosphatase activity towards the HemR. HemR dephosphorylation further leads to differential expression of many genes important in heme metabolism and the inactivation of HemR also generates iron-deficit tolerant phenotype.

For this purpose, they utilized wildtype L. biflexa and generated a double knockout mutant of L. biflexa, ΔhemKR. The approach of engineering the saprophytic strain easier than the virulent strain and it is popular in the field. The study needs few modifications to improve the quality of the manuscript. Below are few recommendations and concerns

1. It would be interesting to see the mechanism in a virulent strain such as L. interrogans. The authors mentioned that the L. biflexa is similar to pathogenic Leptospira species which raises concern as they may look similar but have different membrane proteins which may yield different outcomes during heme metabolism and infection process.

2. The author referred few articles to introduce the heme/iron sensors found across the Leptospira sp. that includes L. interrogans. More recently, Orillard et al in 2022, provided some clear insight about the unusual membrane bound PAS-heme oxygen sensor of L. interrogans. This could be a potential question and use of a pathogenic strain therefore is recommended.

3. The shuttle vector pMaORI is of great choice for this type of study and it creates avenue for many other research questions targeting saprophytic L. biflexa. Is the shuttle vector equally manipulative for pathogenic Leptospira?

4. The author provided evidence that mutant ΔhemKR strain of L. biflexa are more tolerant when there is an iron deficit condition. What is the role of hemKR TCS in the oxidative stress response?

5. What is the role of hemKR TCS in scavenging heme from hemoglobin? This mutant ΔhemKR model may provide better answer and that could be a potential target considering their role during pathogenesis and heme homeostasis.

6. The authors observed a comparatively less significant differences in transcriptional activation and/or down regulation between the wildtype and the mutant strain of L. biflexa and that aligns with the iron deficit response. Although the transcriptomic data during iron overabundance is missing due to some difficulties in RNA purification. May be that would answer whether transcriptional regulation rely on the environmental iron or not.

7. Furthermore, it is not clear from the present study how HemKR responds to ALA and correspond to the environmental iron deficit condition. And, If it does so through transcriptional regulations, how that can influence the iron metabolism during pathogenesis process followed by a virulent Leptospira infection.

8. The author claimed that HemK possesses phosphatase activity in vivo and proposed ALA could be a potential source during iron deficit condition during leptospiral infection in the host. No in vivo results were incorporated to support this.

This study provides a clear insight about the use of a saprophytic leptospiral strain which further creates opportunities for others to address different questions.

Reviewer #2: OVERALL IMPRESSION:

The manuscript entitled “Signal-sensing triggers the shutdown of HemKR, regulating heme and iron metabolism in the spirochete Leptospira biflexa” explores the signaling pathway of the HemKR TCS system that controls heme metabolism and import. To do so, a variety of in vitro approaches were used to carefully answer the questions about the triggering signal of HemK, its effect on activation of its phosphatase activity, the regulatory responses concerning P~HemR and HemR, and, finally, the phenotypical assays that showed that the absence of HemKR control conferred iron-deficit tolerance to leptospires.

The article has the potential to deepen the knowledge on leptospiral regulatory pathways, and metabolism. However, before its publication, the manuscript should be revised to correct a few inaccuracies and to provide additional improvements.

MAJOR ISSUES:

1) On page 16, lines 361-362, it is written: “Transcriptomic analyses confirmed that transcription of hemK and hemR genes was indeed abolished in the mutant strain”. However, a careful examination of Table S2 shows this information is inaccurate. hemR indeed displayed much smaller numbers of read counts when we compare the ∆hemKR strain with the wt, which might indicate a severe impairment in transcription.

However, the two count tables show hemK normalized read counts in the wt and the ∆hemKR strains. Furthermore, when we evaluate the DGE tables, there is differential expression of hemK in ∆hemKR strains: in ALA-treated cultures, hemK was more expressed in comparison to untreated cultures (Log2FC = 0,551; p adj = 1,05E-09). Therefore, if we detect differential expression of hemK, after ALA treatment of ∆hemKR strains, hemK transcription is not abolished in ∆hemKR strains.

I suggest that the authors adapt the aforementioned sentence to better describe the transcriptomic data, since transcription of hemK gene was not abolished.

2) On page 19, lines 417-421, the authors describe a new binding motif of HemR in the upstream region of LEPBIa3430/LEPBIa3431. How was this new motif identified? It is important to include the methods used to detect this binding motif, whether by in vitro or in silico analysis.

MINOR ISSUES:

1) Considering the functional analysis of recombinant TCS proteins, the transcriptomic data of wt and ∆hemKR strains, the binding motif identification, and the phenotypic evaluation of mutant and wt strains under DIP treatment, the authors proposed a signaling and regulatory model of heme metabolism in L. biflexa. I suggest the authors create a schematic figure summarizing the signaling and regulatory pathway of HemKR TCS upon ALA detection, including activated and repressed target genes. Such figure would benefit the readers and highlight your findings.

2) The authors emphasized more than once the hypothetical importance of HemKR TCS system under iron-limiting conditions during host infection by pathogenic Leptospira. Although I agree this is an accurate interpretation of biological evidence, it would be very interesting if the authors could also interpret and discuss their data considering the importance of HemKR system under environmental conditions in water and/or soil. Not only because Leptospira biflexa is a saprophytic species, but because many pathogenic Leptospira spp. present a free-living phase on the environment in their complex life cycle, which enables them to endure between hosts. This would be a very thought-provoking addition.

OTHER POINTS:

Line 55: endoflagella is spelled without a hyphen.

Line 58: “including the causative agent of leptospirosis” should be changed to “aetiological agent of leptospirosis”. A disease is multifactorial and caused by more than its infectious agent, the concentration of bacterial inoculum and the host’s immune system are also unequivocally involved. Therefore, it is not accurate to refer to pathogenic Leptospira as the causative agent of the disease.

6. PLOS authors have the option to publish the peer review history of their article (what does this mean?). If published, this will include your full peer review and any attached files.

Reviewer #1: **Yes: **Suman Kundu

Reviewer #2: No

---

## [Author Response · Author response to Decision Letter 0]

30 Aug 2024

I have uploaded a new Cover Letter responding to the Editor's comments, as well as a point-by-point rebuttal letter (Response to Reviewers) addressing all of the reviewers' concerns.

---

## [Editor Report · Decision Letter 1]

12 Sep 2024

Signal-sensing triggers the shutdown of HemKR, regulating heme and iron metabolism in the spirochete Leptospira biflexa

PONE-D-24-28906R1

Dear Dr. Buschiazzo,

We’re pleased to inform you that your manuscript has been judged scientifically suitable for publication and will be formally accepted for publication once it meets all outstanding technical requirements.

Kind regards,

Brian Stevenson, Ph.D.

Academic Editor

PLOS ONE
---

## [Editor Report · Acceptance letter]

16 Sep 2024

PONE-D-24-28906R1 

PLOS ONE

Dear Dr. Buschiazzo, 

I'm pleased to inform you that your manuscript has been deemed suitable for publication in PLOS ONE. Congratulations! Your manuscript is now being handed over to our production team.

Kind regards, 

on behalf of

Prof. Brian Stevenson 

Academic Editor

PLOS ONE